# Vibrational exercise for Crohn's to observe response (VECTOR): Protocol for a randomized controlled trial

Jonathan Sinclair[1], Johanne Brooks-Warburton[2,3], Lauren Baker[2], Amit N. Pujari[4,5], Matthew Jewiss[6], Charlotte Lawson[7], Simon Anderson[8], Lindsay Bottoms[2*]

1 Research Centre for Applied Sport, Physical Activity and Performance, School of Sport and Health Sciences, Faculty of Allied Health and Wellbeing, University of Central Lancashire, Preston, Lancashire, United Kingdom, 2 Centre for Research in Psychology and Sports, School of Life and Medical Sciences, University of Hertfordshire, Hatfield, Hertfordshire, United Kingdom, 3 East and North Hertfordshire NHS Trust, Stevenage, United Kingdom, 4 Neu(RAL)²: NeuRAL Systems and Rehabilitation and Assistive Technologies Laboratory, School of Physics, Engineering and Computer Science, University of Hertfordshire, Hatfield, Hertfordshire, United Kingdom, 5 School of Engineering, University of Aberdeen, Aberdeen, United Kingdom, 6 The Cambridge Centre for Sport and Exercise Sciences (CCSES), Anglia Ruskin University (ARU), Cambridge, United Kingdom, 7 School of Pharmacy and Biomedical Sciences, University of Central Lancashire, Preston, Lancashire, United Kingdom, 8 Guy's and St Thomas' NHS Foundation Trust, London, United Kingdom

* l.bottoms@herts.ac.uk

## Abstract

Crohn's disease (CD) is a long-term inflammatory gastrointestinal disorder, often adversely affecting physical, emotional, and psychological well-being. Pharmaceutical management is habitually adopted; although medicinal therapies require continuous administration, and are often associated with significant side effects and low adherence rates. Whole body vibration (WBV) represents a non-invasive technique, that provides vibration stimulation to the entire body. As WBV appears to target the physiological pathways and symptoms pertinent to CD epidemiology, it may have significant potential as a novel non-pharmaceutical intervention therapy in CD. This paper presents the study protocol for a randomised controlled trial investigating the impact of WBV on health outcomes in individuals with CD. This 6-week, parallel randomised controlled trial will recruit 168 individuals, assigned to receive WBV and lifestyle education 3 times per week compared to control, receiving lifestyle education only. The primary outcome of the trial will be the difference from baseline to post-intervention in health-related quality of life between the groups, assessed with the Inflammatory Bowel Disease Quality of Life Questionnaire. Secondary outcomes will include between-group differences in other questionnaires assessing fatigue, anxiety and pain, measures of physical fitness, and biological markers for disease activity and inflammation. Statistical analyses will follow an intention-to-treat approach, using linear mixed-effects models to compare changes between time points and both trial groups. Ethical approval was granted by the Nottingham Research Ethics Committee *(REC: 24/EM/0106)* and the study has been registered prospectively as a clinical trial (NTC06211400).

**Data availability statement:** No datasets were generated or analysed during the current study. All relevant data from this study will be made available upon study completion.

**Funding:** Funded fully by Crohn's and Colitis Foundation the funders had no role in study design, data collection and analysis, decision to publish, or preparation of the manuscript.

**Competing interests:** I have read the journal's policy and the authors of this manuscript have the following competing interests: this project is fully funded by an award from Crohn's and Colitis Foundation - award number 1167823. The funder has no role in study design, data collection and analysis, decision to publish or preparation of the manuscript.

## Introduction

Inflammatory bowel diseases (IBD) are long-term disorders that impact the digestive tract [1]. Global incidence of IBD is increasing [2], with its yearly economic burden in the United States surpassing $31 billion [3]. Symptoms associated with IBD can have a profound impact on a person's quality of life, influencing their mental health, body image, social connections, and relationships with friends and family. Moreover, it can lead to decreased participation in the workforce, adding to the financial strain of the disease [4]. The clinical manifestations of IBD vary widely, from asymptomatic or mild phases to severe, life-threatening conditions [1].

The most common forms of IBD are ulcerative colitis (UC) and Crohn's disease (CD) [5]. Specifically, CD can lead to inflammation throughout the digestive tract at any location [5]. Key symptoms of CD include fatigue, abdominal pain, and diarrhoea [6]. Inflammation in the gut wall may impair nutrient absorption, potentially causing reduced bone mineral density and loss of muscle mass. Other non-digestive complications, such as osteoarthritis, sarcopenia, uveitis, iritis, episcleritis, erythema nodosum, and pyoderma gangrenosum, may also arise [6,7]. This broad range of symptoms requires thorough management and support for those affected.

Pharmaceutical intervention is the primary strategy for disease management [8]. However, medicinal therapies for CD usually require continuous administration [9], are associated with significant side effects [10], and are often linked to low adherence rates [11]. Consequently, there is a growing interest in exploring additional treatment approaches for CD [12,13]. Research indicates that up to 56% of IBD patients seek alternative or complementary treatment modalities [14]. This trend highlights the need for more diverse and holistic treatment options to address the complexities of the disease.

Regular physical activity is increasingly recommended as a complementary therapy for managing CD [15]. Intervention studies have shown that low-to-moderate intensity aerobic exercise has several positive effects on individuals with CD, including improving health-related quality of life, reducing inflammation, lessening fatigue, increasing bone mineral density, and enhancing psychological well-being [16–20], all without causing significant adverse effects [21]. Despite these benefits, there is a notable lack of research into the potential advantages of resistance training [22]. Our research revealed that individuals with CD and UC participate far less in resistance exercise compared to aerobic exercise, which appears to stem from a lack of understanding regarding the potential benefits of resistance training in this population especially [12]. Although the precise biological pathways through which exercise exerts its benefits remain unclear, possible explanations include changes in gut microbiota and the influence of physical activity on immune and inflammatory processes [9]. Despite the evidence supporting the positive role of physical activity in CD management, studies have reported that as many as 83% of CD and UC patients are physically inactive [23], with a notable reduction in physical activity levels following diagnosis [24]. The factors contributing to this low engagement in physical activity are not fully understood, with various reasons suggested, including pain and discomfort during exercise, time constraints, fatigue, and fears that physical activity could aggravate symptoms such as abdominal pain and joint issues [12,15,24].

Whole body vibration (WBV) is a non-invasive treatment method that delivers vibrational stimulation to the entire body [25] and can be utilised during resistance exercises like squats. Research in humans has demonstrated that WBV can have beneficial effects on inflammation [26], fatigue [27,28], heart rate variability [29], bone mineral density [30], muscle strength [31], aerobic fitness [32], and pain management [33]. Additionally, studies on animals have indicated that WBV may influence immune cell differentiation and alter the composition of gut microbiota [34]. Alongside the beneficial effects of WBV, one meta-analysis and systematic review including 14 trials, reported no adverse events for WBV exercise[35], whilst another found that only three (17%) studies reported adverse events [36]. These included

redness of the feet, sore feet and headaches, however, it was proposed that these findings were closely related to machine settings and participant set up.

Our previous research showed that vibration exercise superimposed on isometric contractions can generate significantly higher levels of neuromuscular load than static exercise alone [37]. Typically, interventions involving WBV are delivered over shorter durations, 4-6 weeks compared to traditional aerobic exercise interventions, i.e.,10-12 weeks, and each isolated training session is typically of a much shorter duration 10-15 minutes compared to 30 minutes of traditional aerobic exercise. As previously discussed, with time constraints, pain and fatigue cited as significant barriers to physical activity in patients with CD, WBV may represent a more appealing exercise modality. As WBV appears to target physiological pathways [37,38] and symptoms known to be pertinent to CD epidemiology; it may have significant potential as a novel non-pharmaceutical intervention therapy in CD.

While there seems to be a strong rationale for using WBV in therapeutic interventions [39], no trials have yet investigated its effects on CD, and it remains unclear how willing patients are to adopt this potential treatment. Our recent study, which included 463 IBD patients (188 CD patients), showed a high level of interest in trying vibration exercise, indicating significant patient engagement for a randomised clinical trial involving WBV [12]. Furthermore, seeking public and patient involvement, five adults with CD completed 10 minutes of WBV exercise and demonstrated that the approach was well-accepted, with no reported side effects. As a result, further investigation into the potential health benefits of WBV for CD patients could have both practical and clinical significance. This protocol specifically targets individuals with CD, as the pathophysiology differs between UC and CD, and recruiting only CD patients will help minimise heterogeneity in the study population.

The current trial will aim to evaluate the effects of six weeks of WBV (performed three times weekly) on various health outcomes in patients with CD, relative to a control group. The primary objective of the current trial is to assess whether WBV improves health related quality of life (HRQoL) using the Inflammatory Bowel Disease Quality of Life Questionnaire (IBDQ). Secondary outcomes include examining whether WBV exercise can impact an individual's level of fatigue, perception of pain, anxiety and/or depression in CD patients. Questionnaires will include IBD-F for subjective fatigue measures, a pain score questionnaire and HADS for assessment of anxiety/depression. In addition to IBD-F questionnaire, objective muscular fatigue will be assessed via electromyography (EMG). Further secondary outcomes include examining whether WBV can improve overall cardiovascular health, disease activity and markers of inflammation. Here, cardiovascular health will be assessed by the Chester Step Test and blood and faecal samples will determine disease activity and level of inflammation.

It is hypothesised that WBV will lead to significant improvements in self-reported quality of life compared to the control group. Additionally, the researchers anticipate positive changes in patient self-reported fatigue, pain and anxiety/depression, alongside peripheral blood biomarkers indicating reduced inflammation, and physical fitness compared to the control group.

## Methods and materials

### Study design

This study follows protocol version 4 dated 9th of December 2024 (S1 File). The current study complies with the most current guidelines for reporting parallel-group randomised trials [40]. It will be a multi-site, two-arm, individually randomised controlled trial aimed at investigating a specific intervention. The current trial will assess the effectiveness of a 6-week supervised WBV exercise intervention combined with a lifestyle education program, compared to a control group that will receive lifestyle education alone (Fig 1 and Fig 2).

| | STUDY PERIOD | | | |
|---|---|---|---|---|
| | **Enrolment** | **Allocation** | **Post-allocation** | |
| **TIMEPOINT** | **-t1** | **0** | **t₁** | **t₂** |
| | | | **Baseline** | **6-weeks** |
| **ENROLMENT:** | | | | |
| **Eligibility screen** | X | | | |
| **Informed consent** | X | | | |
| **Allocation** | | X | | |
| **INTERVENTIONS:** | | | | |
| Whole body vibration | | | ●————————————● | |
| Control | | | ●————————————● | |
| **ASSESSMENTS:** | | | | |
| ***Blood pressure and resting heart rate*** | | | | |
| Systolic blood pressure | | | X | X |
| Diastolic blood pressure | | | X | X |
| Resting heart rate | | | X | X |
| ***Anthropometrics*** | | | | |
| Body mass | | | X | X |
| ***Fitness testing*** | | | | |
| Physical fitness (examined using the Chester step test) | | | X | X |
| ***Questionnaires*** | | | | |
| Inflammatory Bowel Disease Quality of Life Questionnaire | | | X | X |
| IBD Fatigue Scale | | | X | X |
| Hospital Anxiety and Depression Scale | | | X | X |
| Pain visual analog scale | | | X | X |
| Beck Depression Inventory | | | X | X |
| State Trait Anxiety Inventory | | | X | X |
| ***Heart Rate Variability*** | | | | |
| Standard deviation of NN intervals | | | X | X |
| LF-HRV | | | X | X |
| HF-HRV | | | X | X |
| HF/LF ratio | | | X | X |
| Resting HRV | | | X | X |
| Root mean square of successive RR interval differences | | | | |
| HFnu | | | | |
| ***Faecal samples*** | | | | |
| Faecal calprotectin | | | X | X |
| ***Blood samples*** | | | | |
| Crohn's disease activity index | | | X | X |
| TNF-α | | | X | X |
| IL-6 | | | X | X |
| IL-17A | | | X | X |
| IL-12 | | | X | X |
| IL-23 | | | X | X |
| IL-10 | | | X | X |
| TGF-beta | | | X | X |
| C-Reactive Protein | | | X | X |

**Fig 1. SPIRIT schedule of enrolment, interventions, and assessments.**

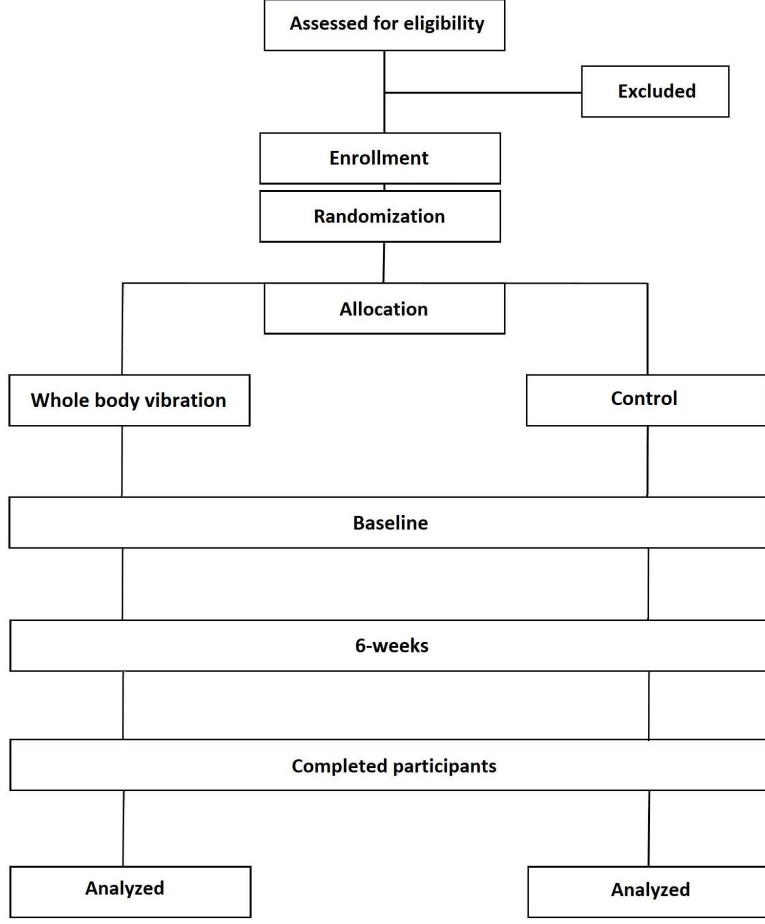

**Fig 2. Consort diagram describing the study design.**

The current trial will be conducted in collaboration with at least two health trusts in England (East and North Hertfordshire NHS Foundation Trust and Bedfordshire Hospitals NHS Trust) and has been adopted by the National Institute for Health Research Clinical Research Network Portfolio, where we are open for additional sites with easy access to Hatfield, England. This will help achieve recruitment target. At each site, participants will be recruited from gastroenterology clinics and all sessions will be delivered by the University of Hertfordshire which is located within the county of Hertfordshire in England.

This study received ethical approval from the Nottingham Research Ethics Committee (REC reference: 24/EM/0106) and is formally registered as a clinical trial on Clinicaltrials.gov (NTC06211400) with the University of Hertfordshire as the Sponsor (research-sponsorship@herts.ac.uk). Any amendments to the protocol will be agreed by the Sponsor prior to being submitted to the REC. The Principal Investigator will update all relevant parties such as the trial registry and Trial Steering Committee.

## Inclusion and exclusion criteria

To qualify for participation in the current trial, individuals must satisfy the following criteria: (1) to be aged between 18 and 65 years, (2) have a confirmed clinical diagnosis of CD for a

minimum of 4 weeks prior to randomization, (3) demonstrate mild to moderate active CD (as indicated by a partial Harvey Bradshaw index (HBI) score of 5-16), (4) have a faecal calprotectin (FC) result < 250 mcg/g recorded no longer than four weeks before randomisation, (5) be on stable medication for at least 4 weeks before randomization, (6) be capable of providing written consent, (7) completing the study questionnaires, and (8) be able to travel to the research centre for assessment visits and exercise sessions.

The exclusion criteria for the current trial are as follows: (1) having any absolute contra-indications to exercise testing and training, such as musculoskeletal injury, (2) the presence of a serious autoimmune disease such as rheumatoid arthritis or systemic sclerosis, (3) any major surgery planned within the first 3 months post-randomization, (4) pregnancy or intentions to become pregnant within the initial 3 months after randomization, (5) poor tolerability to venipuncture or (6) insufficient venous access for necessary blood sampling, (7) involvement in another clinical trial where concurrent participation is considered unsuitable, and (8) any orthopaedic implants (hip, knee, spine). Notably, contra-indications of WBV were identified and included within this criterion and the screening processes for the current trial [41].

## Sample size

As no previous literature exists concerning the effects of WBV in CD, a pragmatic sample size calculation was undertaken using data summarised in a recent meta-analysis [42], exploring the effect of exercise on IBD symptoms. We based our sample size calculation on the mean and standard deviation change in HRQoL from the exercise intervention and control arms of three studies that most closely matched our exercise intervention and utilised the same primary outcome [19,43,44]. Specifically, Cramer et al. [43] reported a mean difference of 24.30 (SD = 34.20) for the exercise group and 6.80 (SD = 17.40) for the control group; Jones et al. [42] reported a mean difference of 4.00 (SD = 10.30) for the exercise group and 0.00 (SD = 12.70) for the control group; and Klare et al. [19] reported a mean difference of 28.30 (SD = 34.50) for the exercise group and 14.50 (SD = 16.10) for the control group. Using these data, the necessary sample size based on the aforementioned data from each previously conducted study was calculated. We then pragmatically calculated the pooled mean of these calculated sample sizes, which showed that to achieve $\alpha$ = 0.05 and $\beta$ = 80%, a total required sample size of 168 participants (84 per group), accounting for an attrition rate of 20% would be necessary. This attrition rate was deemed to be appropriate based on our previous exercise studies in other clinical populations [15,45].

## Participants

Participants included will be CD patients, recruited from gastroenterology clinics at hospitals within commuting distance from the University of Hertfordshire. A member of the clinical team will identify potentially eligible patients using the inclusion/exclusion criteria and provide information regarding the study. Patients will be asked to consent to share contact details with the University of Hertfordshire research team. Once the clinical team have confirmed the patient FC and HBI results are within the accepted range, the patient will be contacted to book in the initial screening/baseline visit to the University of Hertfordshire. At the screening/baseline visit, fully informed consent will be gained, and participants screened and randomised. Online database software, Research Electronic Data Capture (REDCap) will be used to randomly allocate participants to one of two groups for a total duration of six weeks with site stratification. To encourage study adherence all participants will be reimbursed travel expenses for any face to face visits to the University of Hertfordshire.

## Intervention group

For the intervention group, allocated participants will have 18 visits to the University of Hertfordshire for WBV (20 visits in total (+1 if they engage with the semi structured interviews). Participants will be expected to partake in WBV three times a week, for six weeks which could be undertaken at their preferred time of day to increase study adherence. The WBV will involve performing static squatting on a synchronic vibrating platform by (Power Plate, London, United Kingdom). Participants will hold a squatting position with 30˚ of knee flexion, feet about 30 cm apart, barefoot with upper limbs holding the platform bars. Participants will hold this static squat for 60 seconds, then rest for 60 seconds, and repeat six times. The vibratory stimulus will be at an amplitude of 1.5mm with a frequency of 30 Hz. These settings were chosen as earlier findings have shown this amplitude and frequency to be effective in improving muscle activity and strength [31,38,46]. During this visit, participants will receive 10 minutes of the lifestyle education programme, totalling 30 minutes per week. The Lifestyle Education Programme is based around the Crohn's and Colitis UK Patient Information, utilising the Pender Health Promotion Model [47] to identify individual experiences, behaviour specific cognitions and behaviour outcomes.

The specific educational themes within the lifestyle education programme will be as follows:

Week 1 - Healthy Eating (Information from Crohn's and Colitis UK leaflets) Week 2 - Discussing barriers and how to overcome them to Healthy Eating Week 3 - Physical Activity for IBD (Utilizing the Crohn's and Colitis video) Week 4 - Barriers to Physical Activity Week 5 - Overcoming barriers and positive steps forward to engage in Physical Activity. Week 6 - Smoking Cessation utilizing National Health Service England (NHSE) smoking cessation information and signposting to local services if applicable, or a round-up of the last 5 weeks.

## Control group

In the control arm, the aforementioned lifestyle education program will be delivered individually via zoom in 30-minute sessions per week over the six week testing period, using the specific educational themes described above.

## Data collection

Data will be directly entered into REDCap either by the research team or the participant, as an electronic tablet will be provided when at the University of Hertfordshire. All baseline assessments (anthropometrics, blood pressure, resting heart rate, heart rate variability, fitness testing, EMG for assessment of muscular fatigue, questionnaires and blood sampling) will be conducted in Physiology laboratories at the University of Hertfordshire. Participants will then receive either six weeks of WBV and the lifestyle education programme at the University of Hertfordshire, or the lifestyle education programme online if allocated to the control. Following this 6-week period, all assessments conducted at baseline will be repeated at an exit visit for all participants. Due to the nature of the trial, participants will not be blinded from the intervention however the health care professionals and the statistician will be blinded.

## Anthropometrics

Anthropometric data will be collected at both the baseline and exit visit. Body weight (kg) and height (m) (without footwear) will be recorded. Height will be determined using a stadiometer, while weight will be assessed using standard weighing scales.

## Blood pressure and resting heart rate

Systolic and diastolic blood pressure, along with resting heart rate, will be taken at both the baseline and exit visit. This data will be collected using a non-invasive, automated blood pressure monitor, in accordance with the guidelines set by the European Society of Hypertension [48]. Three readings will be recorded, each spaced by a 1-minute interval, with the average of the final two readings adopted for analysis.

## Fitness testing

Cardiovascular fitness level of participants will be explored via the Chester step test at baseline and exit visits [49]. Participants will be fitted with a Polar H10 heart rate monitor (Polar Electro Oy, Kempele, Finland) and step onto either a 15-cm or a 25-cm step at a rate fixed by metronome beats starting with 15 beats per minute. Step height will be determined by physical status, 15 cm for those who take little or no physical exercise and 25 cm for those who regularly take physical exercise. The total test time is 10-minutes comprising of five stages lasting for 2-minutes each and the step rate is increased by five beats for each incremental stage. Participants will rate their perceived exertion using the Borg Scale at the end of every 2-minute stage. Physical fitness will be estimated by plotting each participant's heart rate achieved at the end of each stage on the specified Chester step test data record sheet.

## Electromyography

For each participant, EMG will be recorded at baseline and exit visits, for assessment of muscular fatigue. These EMG recordings will be recorded while participants hold the squat position on the vibration plate, both with and without the vibration stimulation. The EMG will be recorded with active bipolar, wireless EMG electrodes (Biometrics Ltd, Cwmfelinfach, Wales) from two quadriceps muscles (Vastus Lateralis, Rectus Femoris) and two hamstring muscles (Biceps Femoris, Semitendinosus), with a sampling frequency of 1 KHz or higher. The purpose of recording EMG data is to establish the effects of vibration intervention on muscle activation patterns and muscle fatigue.

## Questionnaires

All questionnaires will be completed at both baseline and exit visits for all participants. Firstly, the Inflammatory Bowel Disease Quality of Life Questionnaire (IBD-Q) is widely used and validated for patients with inflammatory bowel disease [50,51]. The IBD-Q consists of 32 items, each rated on a 7-point Likert scale, where higher scores (ranging from 32 to 224) indicate better quality of life related to IBD. It covers four key domains: bowel symptoms, systemic symptoms, emotional well-being, and social functioning. Research has highlighted how the IBD-Q has been widely used as a primary outcome for many clinical trials conducted in this population [52].

Secondly, the IBD-F Scale is another validated tool designed to specifically assess fatigue in individuals with IBD [53]. It includes two parts: the first part assesses overall fatigue levels using five questions, while the second part evaluates the impact of fatigue on quality of life through 30 questions. Each item is scored from 0 to 4, with a maximum total score of 120, where higher scores indicate more severe fatigue and a greater impact on the patient's quality of life. The third questionnaire included is the HADS, which is a validated 14-item scale used to measure symptoms of anxiety and depression. It consists of two subscales—one for anxiety and one for depression, each containing seven items, with individual scores ranging from 0 to 21. Higher scores on either subscale reflect increased levels of anxiety or depression, making it a valuable tool for assessing mental health in this population [54]. Lastly, a visual analogue scale (VAS) will be used to record subjective perceptions of pain. On a scale of 1-10, participants will record their level of

pain associated with their CD symptoms. Pain scores will be recorded for a further 6 days (seven days total), following both the baseline and exit visit to the University of Hertfordshire. Participants will receive a link by email, to complete this daily VAS for pain.

### Heart rate variability

Heart rate variability (HRV) will be assessed for a total of 36 hours (from 7am until 7pm) over a 72-period following the baseline visit, and again for 36 hours after the exit visit. Following both visits, participants will be set up, sent away with the Polar H10 and Polar watch, and provided instructions on use for the 72-hour period. Linear HRV will be recorded using a Polar H10 sensor chest strap device and connected to a Polar Advantage V3 watch (Polar Electro Oy, Kempele, Finland). The purpose of recording HRV is twofold. First as a psychophysiological marker of pain because standard deviation of NN intervals, low frequency (LF)-HRV, high frequency (HF)-HRV and HF/LF Ratio parameters of HRV are acceptable objective indicators of pain [55]. Second as a psychophysiological marker of health and wellbeing because resting HRV (ms), root mean square of successive RR interval differences and HFnu (%) are acceptable markers of healthy functioning and wellbeing [56].

### Blood sampling

Venous blood samples (10 mL) will be drawn from the antecubital vein and immediately placed into VACUETTE serum separator clot activator blood collection tubes (Greiner Bio-One, Kremsmünster, Austria). The collected blood will be left to clot for 30 minutes before separating the serum through centrifugation at $1500 \times g$. The separated serum will be aliquoted and stored at $-80°C$ until further analysis. A custom panel (Legend Plex, Biolegend, London, UK) will be developed for a multiplex assay, which will be carried out using a BD FACS CANTO II flow cytometer (Becton Dickinson, Wokingham, UK) as per the instructions for the Legend Plex setup. This approach will allow us to measure the Crohn's Disease Activity Index [57] and assess both pro-inflammatory cytokines such as TNF-α, IL-6, IL-17A, IL-12/IL-23 and anti-inflammatory cytokines like IL-10 and TGF-beta [58]. C-reactive protein levels will also be assessed, given that it is the most widely used biomarker for inflammation status [59] and exercise has been shown to increase anti-inflammatory cytokines and lower pro-inflammatory ones [60]. In addition, serum P1NP will be measured as an indicator of bone health [61].

### Faecal samples and HBI

Faecal sampling will be conducted by the clinical team at each NHS site. To detect biological variations in the levels of intestinal inflammation, faecal samples will be collected from both groups at the start of the study and again after the 6-week treatment period. These samples will be analysed for faecal calprotectin, which is the most used biomarker for assessing intestinal inflammation and subsequent, disease activity [59,62]. The IBD clinical team responsible will collect samples, and the results will be shared with the research team at University of Hertfordshire for further analysis. Additionally, partial HBI assessment will be carried out by a qualified clinician at each NHS site at the start of the study only. These will be used as part of the inclusion criteria, as they are markers of disease activity.

### Participant semi structured interviews

Fifteen participants from the intervention arm will be invited to undertake a semi structured interview. This will provide granular qualitative data on how they found the intervention,

whether they enjoyed it and whether it caused any discomfort. The interviews will explore whether WBV is an activity they will continue to do, and whether they are likely to have access to WBV following completion of the study. Data will be analysed via thematic analysis [63].

## Safety reporting

Throughout the study, adverse events (AEs) and/or serious adverse events (SAEs) that may occur as a result of participation or lead to study withdrawal will be recorded and documented within REDCap. Procedures for AEs and SAEs outlined within the safety management plan and protocol will be followed, to ensure prompt severity, causality and expectedness grading. Appropriate medical care will be provided by the participant's medical team. Should modifications to the study protocol be necessary, the Trial Steering Committee (TSC) will be notified. The committee will be made up of two members from the Trial Management Group, a statistician, two people with lived experience, a General Practitioner, an exercise specialist, an IBD clinician and will be independent from the funder. If required, data collection will be temporarily halted. Although non-serious adverse events do not meet the criteria for serious medical issues, they will still be thoroughly documented and reviewed. This proactive monitoring process ensures that any potential risks are continuously evaluated and appropriately managed throughout the trial, emphasising participant safety whilst maintaining study integrity. As the trial is a low-risk intervention, the TSC will take on the role and responsibilities for a data monitoring and ethics committee. Central monitoring has been considered appropriate by the Clinical Trials Support Network and each site will have at least one monitoring visit during the trial. An audit visit may be conducted by the Sponsor at their request and is not required by the funder.

## Data management

Data collection and storage procedures will comply with the Data Protection Act 2018. Data entry, including electronic consent, will be managed using the REDCap system, ensuring confidentiality and accuracy. Participant information will be anonymised for analysis. After study completion, all data will be transferred to the University of Hertfordshire's Research Data Archive (UHRA), where they will be securely stored for seven years, following legal and ethical guidelines. During this period, the data will remain protected and accessible only to authorised personnel such as the trial manager and trial statistician. After seven years, the data will be disposed of according to the university's data management policies, ensuring proper confidentiality and security.

## Statistical analysis

For continuous measurements, means and standard deviations will be reported. The analysis for the intervention will follow an intention-to-treat method. To evaluate the effect of the intervention on the outcome measures, we will compare the changes observed from baseline to the 6-week follow-up between the two groups. This comparison will be conducted using linear mixed-effects models, employing restricted maximum likelihood estimation and compound symmetry techniques. Trial group assignment will be treated as a fixed factor, with random intercepts assigned to individual participants [64]. The effect sizes for differences between groups over the 6-week period will be calculated using Cohen's d, based on the guidelines of McGough and Faraone [65], where a value of 0.2 represents a small effect, 0.5 indicates a moderate effect, and 0.8 denotes a large effect [66]. In accordance with recommendations [67], missing data will be inputted using multiple imputation, employing a fully conditional specification approach. All statistical analyses will be performed using SPSS software version 29 (IBM Inc., SPSS, Chicago, IL, USA), and statistical significance will be defined as $P \leq 0.05$.

### Timelines

Recruitment will commence in February of 2025 until September of 2026.The proposed trial end date is September 2027.

## Discussion

### Expected results

Anticipated results of this study are expected to demonstrate notable enhancements in self-reported quality of life for participants engaging in the WBV intervention when compared to the control group. Furthermore, we foresee favourable outcomes in various health-related questionnaires, a reduction in inflammation as reflected by peripheral blood biomarkers. Should these hypotheses be validated, the findings could provide valuable clinical insights. Considering the challenging nature of CD, along with its significant healthcare costs and detrimental effects on quality of life and mental health, these outcomes could have substantial implications for improving the management and treatment of CD through the incorporation of WBV interventions. The potential benefits could lead to enhanced patient well-being and more effective therapeutic strategies.

### Dissemination

Following data analysis, participants will receive a newsletter summarising study outcomes. The main dissemination of the trial findings will be through publications in peer-reviewed academic journals and presentations at both national and international conferences. As part of a comprehensive dissemination strategy, study results will be communicated to a broad audience at local, national, and international levels. A key feature of this strategy includes the involvement of a CD patient in the Trial Steering Committee, ensuring patient representation and participation in decision-making throughout the trial. Once the study concludes, findings will be shared with local IBD support groups. To extend the reach of this information, we will collaborate with patient representatives and regional authorities on patient and public engagement initiatives. A detailed informational article summarising study outcomes will be distributed to relevant organizations. Furthermore, peer-reviewed journal articles will be produced and presentations delivered at international conferences. These steps will guarantee results reach a wide range of stakeholders, including healthcare professionals, policymakers, and the patient community.

## Supporting information

**S1 File. Protocol V4.**
(PDF)

## Author contributions

**Conceptualization:** Lindsay Bottoms, Johanne Brooks-Warburton, Amit N. Pujari, Matthew Jewiss, Charlotte Lawson, Simon Anderson.

**Funding acquisition:** Lindsay Bottoms, Jonathan Sinclair, Johanne Brooks-Warburton, Amit N. Pujari, Matthew Jewiss, Charlotte Lawson, Simon Anderson.

**Methodology:** Lindsay Bottoms, Jonathan Sinclair, Johanne Brooks-Warburton, Lauren Baker, Amit N. Pujari, Matthew Jewiss, Charlotte Lawson, Simon Anderson.

**Project administration:** Johanne Brooks-Warburton, Lauren Baker.

**Writing – original draft:** Lindsay Bottoms, Jonathan Sinclair, Johanne Brooks-Warburton, Lauren Baker, Amit N. Pujari, Matthew Jewiss, Charlotte Lawson, Simon Anderson.

**Writing – review & editing:** Lindsay Bottoms, Jonathan Sinclair, Johanne Brooks-Warburton, Lauren Baker, Amit N. Pujari, Matthew Jewiss, Charlotte Lawson, Simon Anderson.

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
