## [Decision Letter · Decision Letter 0]

19 Dec 2024

PONE-D-24-48588Vibrational Exercise for Crohn’s to Observe Response (VECTOR): Protocol for a Randomized Controlled Trial.PLOS ONE

Dear Dr. Bottoms,

Thank you for submitting your manuscript to PLOS ONE. After careful consideration, we feel that it has merit but does not fully meet PLOS ONE’s publication criteria as it currently stands. Therefore, we invite you to submit a revised version of the manuscript that addresses the points raised during the review process.

We look forward to receiving your revised manuscript.

Kind regards,

Amna Aamir Khan, PhD, MScPT, ADPT, BSPT

Academic Editor

PLOS ONE

Journal Requirements: When submitting your revision, we need you to address these additional requirements. 1. Please ensure that your manuscript meets PLOS ONE's style requirements, including those for file naming. The PLOS ONE style templates can be found at https://journals.plos.org/plosone/s/file?id=wjVg/PLOSOne_formatting_sample_main_body.pdf and https://journals.plos.org/plosone/s/file?id=ba62/PLOSOne_formatting_sample_title_authors_affiliations.pdf 2. We note that the grant information you provided in the ‘Funding Information’ and ‘Financial Disclosure’ sections do not match.  When you resubmit, please ensure that you provide the correct grant numbers for the awards you received for your study in the ‘Funding Information’ section. 3. Thank you for stating the following financial disclosure: "Funded fully by Crohn's and Colitis Foundation" Please state what role the funders took in the study.  If the funders had no role, please state: ""The funders had no role in study design, data collection and analysis, decision to publish, or preparation of the manuscript."" If this statement is not correct you must amend it as needed. Please include this amended Role of Funder statement in your cover letter; we will change the online submission form on your behalf. 4. Please include captions for your Supporting Information files at the end of your manuscript, and update any in-text citations to match accordingly. Please see our Supporting Information guidelines for more information: http://journals.plos.org/plosone/s/supporting-information. 5. We note that the original protocol that you have uploaded as a Supporting Information file contains an institutional logo. As this logo is likely copyrighted, we ask that you please remove it from this file and upload an updated version upon resubmission.

**Additional Editor Comments:**

Editors comments

Subheadings such as "Aims & Objectives," "Rationale," and "Hypotheses" are unnecessary and should be removed.

The term "health-related questionnaires" is vague and insufficient, requiring more elaboration.

Introduction

• The term "Inflammatory bowel diseases" is overly emphasized, while the primary sample, "Crohn's disease," is underrepresented.

Methods

• The methods section is overly complex and filled with unnecessary details, yet critical elements are missing.

• Ethical approval, a fundamental requirement, was not mentioned in right place, which is a significant oversight.

• Exclusion criteria are inadequate. For instance, conditions like knee pain are not addressed, and related protocols remain unclear.

• Secondary outcomes, such as stool biomarkers and VAS for pain assessment, are insufficiently detailed and lack clarity.

In conclusion, your manuscript requires substantial revisions in both scientific content and writing quality. It cannot be accepted in its current state.

Reviewer 1 Comments:

The study can not be blinded to the participants. How will the potential bias be handled?

Sample size calculation needs more details. What are the assumed mean and standard deviation? What test was used?

For the linear mixed model, may consider other covariance matrices in addition to CS.

How will missing data and early termination be handled?

Reviewer 2 Comments:

) Include a brief discussion on potential adverse effects or contraindications of WBV, supported by references.

2) The WBV protocol description is clear but lacks justification for the chosen frequency and amplitude settings. (method section) Provide a brief explanation or reference supporting the selection of the vibration parameters (1.5 mm, 30 Hz).

3) The rationale for selecting specific biomarkers and quality-of-life questionnaires is not fully explained.

4) Justify the inclusion of certain biomarkers (e.g., calprotectin, cytokines) and how they specifically reflect WBV effects.

5) Provide brief captions or explanations for Figure 1 and Figure 2 within the manuscript.

Reviewers' comments:

Reviewer's Responses to Questions

**Comments to the Author**

1. Does the manuscript provide a valid rationale for the proposed study, with clearly identified and justified research questions?

Reviewer #1: Yes

Reviewer #2: Yes

Reviewer #3: Yes

2. Is the protocol technically sound and planned in a manner that will lead to a meaningful outcome and allow testing the stated hypotheses?

Reviewer #1: Partly

Reviewer #2: Yes

Reviewer #3: Partly

3. Is the methodology feasible and described in sufficient detail to allow the work to be replicable?

Reviewer #1: No

Reviewer #2: Yes

Reviewer #3: Yes

4. Have the authors described where all data underlying the findings will be made available when the study is complete?

Reviewer #1: No

Reviewer #2: Yes

Reviewer #3: Yes

5. Is the manuscript presented in an intelligible fashion and written in standard English?

Reviewer #1: No

Reviewer #2: Yes

Reviewer #3: Yes

6. Review Comments to the Author

You may also provide optional suggestions and comments to authors that they might find helpful in planning their study.

Reviewer #1: Thank you for your submission. While your study aims to provide an innovative approach to managing Crohn's disease, it has been found to be insufficient in terms of scientific standards and writing quality. Therefore, your manuscript cannot be accepted for publication in its current form. Below are the reasons for this decision:

General Assessment

• The methodological structure and writing style of your manuscript do not meet the requirements of a scientific publication.

• Significant gaps and inconsistencies are observed in both the introduction and methods sections.

Abstract

• The abstract is overly generic and does not clearly reflect the key points of the study.

• Details about the intervention for the control group are missing.

• The term "health-related questionnaires" is vague and insufficient, requiring more elaboration.

Introduction

• The term "Inflammatory bowel diseases" is overly emphasized, while the primary sample, "Crohn's disease," is underrepresented.

• Subheadings such as "Aims & Objectives," "Rationale," and "Hypotheses" are unnecessary and should be removed.

• Unpublished and small-sample pilot studies should not be presented as evidence but rather with caution.

• The writing style lacks academic rigor and needs to adopt a more professional and formal tone.

Methods

• The methods section is overly complex and filled with unnecessary details, yet critical elements are missing.

• Ethical approval, a fundamental requirement, was not mentioned in right place, which is a significant oversight.

• Exclusion criteria are inadequate. For instance, conditions like knee pain are not addressed, and related protocols remain unclear.

• Secondary outcomes, such as stool biomarkers and VAS for pain assessment, are insufficiently detailed and lack clarity.

Outcome Measures

• The primary and secondary outcome measures lack clear criteria. The focus should shift towards more clinical and measurable outcomes.

Writing and Formatting

• The overall writing style falls short of academic standards. Excessive use of subheadings and overly detailed descriptions result in a lack of focus.

• Headings and subheadings are inconsistent and need careful revision throughout the manuscript.

In conclusion, your manuscript requires substantial revisions in both scientific content and writing quality. It cannot be accepted in its current state.

Reviewer #2: The study can not be blinded to the participants. How will the potential bias be handled?

Sample size calculation needs more details. What are the assumed mean and standard deviation? What test was used?

For the linear mixed model, may consider other covariance matrices in addition to CS.

How will missing data and early termination be handled?

Reviewer #3: 1) Include a brief discussion on potential adverse effects or contraindications of WBV, supported by references.

2) The WBV protocol description is clear but lacks justification for the chosen frequency and amplitude settings. (method section) Provide a brief explanation or reference supporting the selection of the vibration parameters (1.5 mm, 30 Hz).

3) The rationale for selecting specific biomarkers and quality-of-life questionnaires is not fully explained.

4) Justify the inclusion of certain biomarkers (e.g., calprotectin, cytokines) and how they specifically reflect WBV effects.

5) Provide brief captions or explanations for Figure 1 and Figure 2 within the manuscript.

7. PLOS authors have the option to publish the peer review history of their article (what does this mean? ). If published, this will include your full peer review and any attached files.

**Do you want your identity to be public for this peer review?** For information about this choice, including consent withdrawal, please see our Privacy Policy .

Reviewer #1: No

Reviewer #2: No

Reviewer #3: **Yes: ** Esedullah AKARAS

---

## [Author Response · Author response to Decision Letter 1]

30 Jan 2025

Thank you very much to the Editor and all the Reviewers for your comments. Please see below our responses to the comments. We have signposted where the amendments have been made by specifying which line, in response to your comments below.

To confirm, we have submitted;

Vibration_protocol paper_REVISED MANUSCRIPT WITH TRACK CHANGES

Vibration_protocol paper_REVISED FINAL

Thank you very much.

and

Response: Thank you, we have ensured the manuscript meets the requirements above and amended appropriately.

Response: Amended these sections and checked grant numbers.

3. Thank you for stating the following financial disclosure: "Funded fully by Crohn's and Colitis Foundation". Please state what role the funders took in the study. If the funders had no role, please state: ""The funders had no role in study design, data collection and analysis, decision to publish, or preparation of the manuscript.""

If this statement is not correct you must amend it as needed. Please include this amended Role of Funder statement in your cover letter; we will change the online submission form on your behalf.

Response: Thank you very much, amended both.

Response: Thank you for your comment. Unless we’re misunderstanding, we do not have any supporting information to provide at the end of the manuscript. Our figures are not supposed to be supporting information, thank you.

5. We note that the original protocol that you have uploaded as a Supporting Information file contains an institutional logo. As this logo is likely copyrighted, we ask that you please remove it from this file and upload an updated version upon resubmission.

Response: Logo removed, and protocol updated upon resubmission.

Editor Comments:

Subheadings such as "Aims & Objectives," "Rationale," and "Hypotheses" are unnecessary and should be removed.

Response: Thank you for your comment, these have been removed.

The term "health-related questionnaires" is vague and insufficient, requiring more elaboration.

Response: Thank you for your comment, more elaboration on the secondary goals and questionnaires used has been provided in line 154-157.

Introduction

• The term "Inflammatory bowel diseases" is overly emphasized, while the primary sample, "Crohn's disease," is underrepresented.

Response: Thank you for your comment, the authors agree and have edited the introduction so that Crohn’s (CD) is more clearly referred to throughout. The initial paragraph has been re-structured to speak about IBD generally first, then narrow down to CD and the intervention rational.

Methods

• The methods section is overly complex and filled with unnecessary details, yet critical elements are missing.

Response: Thank you for your comment. We believe that we have now made the methods more succinct, deleting and amending sub-headings. With this, we hope we’ve made the protocol easier to follow and thus it should be replicable. We have added EMG and more clarity of the FC/HBI sections within the methods.

• Ethical approval, a fundamental requirement, was not mentioned in right place, which is a significant oversight.

Response: Thank you, apologies for this. Line 187 - Ethical approval has been mentioned earlier, in the correct section.

• Exclusion criteria are inadequate. For instance, conditions like knee pain are not addressed, and related protocols remain unclear.

Response: Thank you for your comment, more detail has been added into the exclusion criteria. Please note, another point has been added to exclusion to address musculoskeletal injury as queried here, line 204.

• Secondary outcomes, such as stool biomarkers and VAS for pain assessment, are insufficiently detailed and lack clarity.

Response: Thank you for your comment, more detail has been added on secondary outcomes and how they will be assessed (line 152). We hope that more clarity on these two data points is provided within the methods section (FC, line 377; VAS for pain, line 339).

In conclusion, your manuscript requires substantial revisions in both scientific content and writing quality. It cannot be accepted in its current state.

Response: Thank you very much for your comment, we have revised the manuscript and believe we have amended appropriately for publication.

Reviewer 1 Comments:

• The study can not be blinded to the participants. How will the potential bias be handled?

Response: Thank you for your comment. We are trying to match contact time between the two groups, with the lifestyle education programme but unfortunately, one limitation is that we cannot eliminate this bias.

• Sample size calculation needs more details. What are the assumed mean and standard deviation? What test was used?

Response: Thank you for your comment. There is no existing or pilot data examining WBV on our primary outcome or indeed any other measure as far as we are aware in IBD, to enable the authors to generate a mean and SD value. To perform a sample size estimation, a pragmatic calculation was undertaken. This was done based on data from the Cramer, Jones and Klare studies cited in our paper already, more detail has been added in line 221.

• For the linear mixed model, may consider other covariance matrices in addition to CS.

Response: Thank you very much for your comment, we will consider this. We understand it is straightforward to do in the statistical software, but at study commencement, it was deemed that adjustment for covariate variables in the LMM was not necessary.

• How will missing data and early termination be handled?

Response: Thank you for your comment. Participants drop out recording and analysis has already been described in the paper. However, missing data will be handled using methods described in line 444.

Reviewer 2 Comments:

• Include a brief discussion on potential adverse effects or contraindications of WBV, supported by references- add some in about no contraindications.

Response: Thank you for your comment, a brief discussion on potential adverse effects of WBV has been added into the introduction (line 116). Furthermore, a sentence has been added into the exclusion criteria confirming that contraindications of WBV (with reference) were considered for the creation for participant exclusion and screening processes.

• The WBV protocol description is clear but lacks justification for the chosen frequency and amplitude settings (method section). Provide a brief explanation or reference supporting the selection of the vibration parameters (1.5mm, 30 Hz)

Response: Thank you for your comment, a brief explanation with references, supporting the selection of the vibration parameters is now included (line 386).

• The rationale for selecting specific biomarkers and quality-of-life questionnaires is not fully explained

Response: Thank you for your comment. The IBD-Q is gold standard, and justification has been provided in line 326. The reference highlights that many clinical trials use this questionnaire as a primary outcome. In regard to blood biomarkers, CRP is the most widely used biomarker for inflammation, and this has been explained in line 372. As WBV has been shown to reduce inflammation, this is justification of including this biomarker.

• Justify the inclusion of certain biomarkers (e.g., calprotectin, cytokines) and how they specifically reflect WBV effects.

Response: Thank you very much for your comment. Faecal calprotectin does not specifically reflect WBV effects; this is a marker of Crohns disease activity which was explained in line. Justification for this is to determine whether the WBV has any influence on disease activity and associated inflammation. Regarding cytokines, we believe these markers are more exploratory.

• Provide brief captions or explanations for Figure 1 and Figure 2 within the manuscript.

Response: Thank you for your comment. The current authors believe that explanations are not required for Figures 1 and 2 within the manuscript. They refer to required documentation for submission and do not require an explanation.

Reviewer #1: Thank you for your submission. While your study aims to provide an innovative approach to managing Crohn's disease, it has been found to be insufficient in terms of scientific standards and writing quality. Therefore, your manuscript cannot be accepted for publication in its current form. Below are the reasons for this decision:

General Assessment

• The methodological structure and writing style of your manuscript do not meet the requirements of a scientific publication.

Response: Thank you for your assessment, this has been amended.

• Significant gaps and inconsistencies are observed in both the introduction and methods sections.

Response: Thank you for your comment. This has also been amended; gaps and inconsistencies have been addressed in both sections.

Abstract

• The abstract is overly generic and does not clearly reflect the key points of the study.

Response: Thank you for your comment, we believe this has now been addressed. More detail has been added to reflect the key outcomes for the study.

• Details about the intervention for the control group are missing.

Response: Thank you, this information has been added.

• The term "health-related questionnaires" is vague and insufficient, requiring more elaboration.

Response: Thank you for your comment, this term has been removed throughout and elaborated upon.

Introduction

• The term "Inflammatory bowel diseases" is overly emphasized, while the primary sample, "Crohn's disease," is underrepresented.

Response: Thank you for your comment, we have amended and referred to the target population group of CD where applicable.

• Subheadings such as "Aims & Objectives," "Rationale," and "Hypotheses" are unnecessary and should be removed.

Response: Thank you for your comment, these subheadings have been removed.

• Unpublished and small-sample pilot studies should not be presented as evidence but rather with caution.

Response: Thank you for your comment. This has been amended so it is more clearly stated as patient and public involvement work, to aid the set up of this current clinical trial.

• The writing style lacks academic rigor and needs to adopt a more professional and formal tone.

Response: Thank you for your comment, we believe this has now been addressed throughout.

Methods

• The methods section is overly complex and filled with unnecessary details, yet critical elements are missing.

Response: Thank you for your comment. We believe that we have now made the methods more succinct, deleting and amending sub-headings. With this, we hope we’ve made the protocol easier to follow and thus replicable.

• Ethical approval, a fundamental requirement, was not mentioned in right place, which is a significant oversight.

Response: Thank you for your comment, apologies for this. Moved and amended.

• Exclusion criteria are inadequate. For instance, conditions like knee pain are not addressed, and related protocols remain unclear.

Response: Added extra information and comment addressed, thank you.

• Secondary outcomes, such as stool biomarkers and VAS for pain assessment, are insufficiently detailed and lack clarity.

Response: Thank you for your comment, more detail has been added on secondary outcomes and how they will be assessed (line 152). We hope that more clarity on these two data points is provided within the methods section (FC, line 377; VAS for pain, line 339).

Outcome Measures

• The primary and secondary outcome measures lack clear criteria. The focus should shift towards more clinical and measurable outcomes.

Response: Thank you for your comment. We hope more clarity has been provided in the primary and secondary outcomes section (lines 148 onwards). We understand that our primary outcome is symptom focussed due to the IBD-Q, but we have included more clinical and measurable secondary outcomes. A core outcome recommended includes a mixture of self-reported symptoms (subjective) and clinical biomarkers (objective), which we believe has been done.

Writing and Formatting

• The overall writing style falls short of academic standards. Excessive use of subheadings and overly detailed descriptions result in a lack of focus.

Response: Thank you for your comment, subheadings have been deleted to improve manuscript flow.

• Headings and subheadings are inconsistent and need careful revision throughout the manuscript.

Response: This has been amended, thank you.

In conclusion, your manuscript requires substantial revisions in both scientific content and writing quality. It cannot be accepted in its current state.

Response: Thank you for taking the time to read our manuscript, we hope the amendments have improved this for publication.

Reviewer #2:

• The study can not be blinded to the participants. How will the potential bias be handled?

• Sample size calculation needs more details. What are the assumed mean and standard deviation? What test was used?

• For the linear mixed model, may consider other covariance matrices in addition to CS.

• How will missing data and early termination be handled?

Reviewer #3:

• Include a brief discussion on potential adverse effects or contraindications of WBV, supported by references.

• The WBV protocol description is clear but lacks justification for the chosen frequency and amplitude settings (method section) Provide a brief explanation or reference supporting the selection of the vibration parameters (1.5mm, 30 Hz).

• The rationale for selecting specific biomarkers and quality-of-life questionnaires is not fully explained.

• Justify the inclusion of certain biomarkers (e.g., calprotectin, cytokines) and how they specifically reflect WBV effects.

• Provide brief captions or explanations for Figure 1 and Figure 2 within the manuscript.

Response: Thank you very much. The above comments by reviewer 2 and 3 were repeated and have already been addressed.

Other notable changes by Authors:

Line 171- Randomisation method has been changed

Line 190- Added in the third NHS trust

Line 202- Another inclusion criteria point added

Line 307/321/361/372- Detailed information about what equipment and consumables will be used, in the hope that this makes the protocol more replicable (as stated in one of the reviewer questions).

Line 488- The authors have decided to delete one paragraph from the expected results section, as this was simply repeating aims rather than adding to this section.

---

## [Decision Letter · Decision Letter 1]

6 Feb 2025

Vibrational Exercise for Crohn’s to Observe Response (VECTOR): Protocol for a Randomized Controlled Trial.

PONE-D-24-48588R1

Dear Dr. Bottoms,

We’re pleased to inform you that your manuscript has been judged scientifically suitable for publication and will be formally accepted for publication once it meets all outstanding technical requirements.

Kind regards,

Amna Aamir Khan, PhD, MScPT, ADPT, BSPT

Academic Editor

PLOS ONE

Additional Editor Comments (optional):

Reviewers' comments:

Reviewer's Responses to Questions

**Comments to the Author**

1. Does the manuscript provide a valid rationale for the proposed study, with clearly identified and justified research questions?

Reviewer #2: Yes

2. Is the protocol technically sound and planned in a manner that will lead to a meaningful outcome and allow testing the stated hypotheses?

Reviewer #2: Yes

3. Is the methodology feasible and described in sufficient detail to allow the work to be replicable?

Reviewer #2: Yes

4. Have the authors described where all data underlying the findings will be made available when the study is complete?

Reviewer #2: Yes

5. Is the manuscript presented in an intelligible fashion and written in standard English?

Reviewer #2: Yes

6. Review Comments to the Author

You may also provide optional suggestions and comments to authors that they might find helpful in planning their study.

Reviewer #2: All my concerns are addressed.

7. PLOS authors have the option to publish the peer review history of their article (what does this mean? ). If published, this will include your full peer review and any attached files.

**Do you want your identity to be public for this peer review?** For information about this choice, including consent withdrawal, please see our Privacy Policy .

Reviewer #2: No

---

## [Editor Report · Acceptance letter]

PONE-D-24-48588R1

PLOS ONE

Dear Dr. Bottoms,

I'm pleased to inform you that your manuscript has been deemed suitable for publication in PLOS ONE. Congratulations! Your manuscript is now being handed over to our production team.

Kind regards,

on behalf of

Dr. Amna Aamir Khan

Academic Editor

PLOS ONE